Molecular response of canola to salt stress: insights on tolerance mechanisms

Shokri-Gharelo Reza rezashokri92@ms.tabrizu.ac.ir shokri.gharelo@gmail.com 1
Noparvar Pouya Motie 1 2
1 Department of Plant Breeding and Biotechnology, University of Tabriz , Tabriz , Iran
2 Young Researchers and Elite Club, Islamic Azad University , Tabriz , Iran
Uversky Vladimir
Electronic publication date: 2018 May 22
Publication date: 2018
Volume: 6
Electronic Location ID: e4822
Received 2018 Mar 9; Accepted 2018 May 2
Copyright: ©2018 Shokri-Gharelo and Noparvar
Copyright year: 2018
Copyright holder: Shokri-Gharelo and Noparvar
License: This is an open access article distributed under the terms of the Creative Commons Attribution License, which permits unrestricted use, distribution, reproduction and adaptation in any medium and for any purpose provided that it is properly attributed. For attribution, the original author(s), title, publication source (PeerJ) and either DOI or URL of the article must be cited.
License URL: https://creativecommons.org/licenses/by/4.0/

Keywords: Epigenetic Modifications, miRNA, Gene Regulation, NaCl, Proteomics

Funding: The authors received no funding for this work.

==============================
Canola (Brassica napus L.) is widely cultivated around the world for the production of edible oils and biodiesel fuel. Despite many canola varieties being described as ‘salt-tolerant’, plant yield and growth decline drastically with increasing salinity. Although many studies have resulted in better understanding of the many important salt-response mechanisms that control salt signaling in plants, detoxification of ions, and synthesis of protective metabolites, the engineering of salt-tolerant crops has only progressed slowly. Genetic engineering has been considered as an efficient method for improving the salt tolerance of canola but there are many unknown or little-known aspects regarding canola response to salinity stress at the cellular and molecular level. In order to develop highly salt-tolerant canola, it is essential to improve knowledge of the salt-tolerance mechanisms, especially the key components of the plant salt-response network. In this review, we focus on studies of the molecular response of canola to salinity to unravel the different pieces of the salt response puzzle. The paper includes a comprehensive review of the latest studies, particularly of proteomic and transcriptomic analysis, including the most recently identified canola tolerance components under salt stress, and suggests what researchers should focus on in future studies.

Introduction

Salinity is one of the most important environmental factors that affect the distribution and abundance of plant species. Soil salinization occurs mainly in two ways: high evaporation relative to precipitation in association with weak leaching in soils, and salt accumulation as a result of the use of saline water (Singh, 2015). It is estimated that about 50% of the world’s land will be saline by the middle of the 21st century (Mahajan & Tuteja, 2005).

Soils with high levels of salinity have a low water potential zone; consequently, it is difficult for the plant to absorb water and nutrients. In other words saline soils expose plants to osmotic stress (Agarwal et al., 2013). One of the most important consequences of osmotic stress on plants is the production of reactive oxygen species (ROS) in large amounts that followed by oxidative damages, e.g., the degradation of proteins, lipids, pigments, and DNA (Das & Roychoudhury, 2014). Plants growing on saline conditions take up harmful ions, especially Na+ and Cl−ions. Accumulation of Na+ and Cl− ions in large amounts is toxic for the cell, and compounds osmotic stress (Agarwal et al., 2013). These ions disrupt membrane integrity, cell metabolism, enzyme structure, cell growth, and photosynthesis (Ghosh & Xu, 2014).

Although plants have a variety of ways of withstanding the stress, significant loss of yield occurs (Deinlein et al., 2014). Meanwhile, there is an increasing need to produce enough food for the world’s growing population (Deinlein et al., 2014; Singh, 2015). In order to address these challenges to the world’s food security, the engineering of plants to create species that tolerate salinity has been considered as a promising strategy. Achieving salt-tolerance in plants requires comprehensive knowledge of plant molecular mechanisms behind salt tolerance.

Canola (Brassica napus L.) belongs to the genus Brassica from the family Brassicaceae. Canola, also called rapeseed, is the third most important crop after palm and soybean, cultivated worldwide for oil production, and considered as one of the essential sources for biodiesel fuel (Carré & Pouzet, 2014; Milazzo et al., 2013). Like other important crops, salt stress reduces canola yield and production. Some canola cultivars show high tolerance to salinity, while others are susceptible (Banaei-Asl et al., 2015; Banaei-Asl et al., 2016; Bandehagh et al., 2011; Gharelo-Shokri et al., 2016). In spite of extensive studies of canola cultivars under salt stress (Table 1), progress in engineering tolerant canola has been slow, mainly because of the complexity of the molecular mechanisms involved and the lack of sufficient information.

Table 1 Recent studies identifying some of canola salt-tolerant components at the molecular level, including genes, proteins, miRNAs, epigenetic modifications, enzyme activities, and metabolic pathways.

No.	Cultivar	Tissue	Salt treatment	Method	Reference	
1	Zhongyou 821	Root, shoot, and leaf	250 and 400 µM of mannitol	RT-PCR	Zhong et al. (2012)	
2	Nannongyou No.3	Leaf	200 mM NaCl	2-DE	Jia et al. (2015)	
3	Westar	Plantlet	10, 20, 50, 100, 150, 200, 300, 400, 500, 750 and 1,000 mM NaCl	AFLP assay	Guangyuan et al. (2007)	
4	Dunkled, CON-III, Rainbow, Cyclone, Hyola 308, Hyola 401, Hyola 60, Optlon 50 and RGS003	Root, shoot	150, 200 and 300 mM NaCl	Physiological characteristics	Ashraf & Ali (2008) and Heidari (2010)	
5	Hyola 308, Sarigol	Leaf	175 and 350 mM NaCl	2-DE	Bandehagh et al. (2011)	
6	Hyola 308, Sarigol	Root, leaf	150 and 300 mM NaCl	Gel-free proteomics	Banaei-Asl et al. (2015) and Banaei-Asl et al. (2016)	
7	N119	Root, leaf	200 mM NaCl	RNA-seq	Yong et al. (2014)	
8	Sary	Leaf	150 mM NaCl	2-DE	Yíldíz, Akçalí & Terzi (2015)	
9	Hyola 308, Sarigol	Root, leaf	300 mM NaCl	In silico	Gharelo & Bandehagh (2017)	
10	ZS11	Seed	200 mM NaCl	qRT-PCR	Jian et al. (2016)	
11	Westar	Leaf	75 and 150 mM NaCl	Glutathione synthesis assay	Ruiz & Blumwald (2002)	
12	Dunkled, Cyclone	Leaf	150 mM NaCl	RT-PCR	Saadia et al. (2012)	
13	Lines 2205 and 1423	Root, leaf	100 and 200 mM NaCl	QTL mapping	Lang et al. (2017)	
14	Chiifu	Whole plant	250 mM NaCl	Microarray	Lee et al. (2008)	
15	Westar	Seed, young plant, leaf, bud, flower, and root	300 mM NaCl	Genetic engineering	Das et al. (2005)	
Notes.

Abbreviations 2-DE Two-dimensional gel electrophoresis

AFLP assay Amplified fragment length polymorphism

qRT-PCR Quantitative real-time polymerase chain reaction

QTL mapping Quantitative trait locus

RT-PCR Real-time polymerase chain reaction

Several review articles have been written, mostly about the morphological, physiological, and biochemical response of canola to salt stress (Ashraf & McNeilly, 2004; Kumar et al., 2015; Zhang et al., 2014b). Canola cultivars respond to salinity by changes in their morphological, physiological, and biochemical characteristics as well as molecular changes. The relationship of canola cultivars with different ploidy levels and their cell/whole-plant level response to salinity have been reviewed and a number of factors contributing to tolerance have been summarized (Ashraf & McNeilly, 2004). In another review, Zhang et al. (2014b) focused mainly on canola response to salinity in terms of physiological response, classical genetics and QTL mapping. Finally, Kumar et al. (2015) reviewed the molecular breeding of canola to salt stress with the main focus on molecular markers. However, very little attention has been paid to components of the salinity response molecular network—essential for engineering salt-tolerant canola.

In this paper, we focus on studies that aim to illustrate canola molecular mechanism(s) under salt stress using proteomics, transcriptomics, and genetic engineering methods. Our review aims to improve understanding of the known aspects of the response of canola to salinity, identify unknown or less-known aspects of this response, and identify tolerance mechanisms at the molecular level.

Survey methodologys

We searched literature relevant to the topic of the article using Google Scholar and PubMed. Key words such as “canola,” “Brassica,” “salt stress,” “salinity,” “tolerance mechanism,” “proteomic analysis,” “transcriptomic analysis,” plant salt-tolerance mechanism,” “gene regulation,” “proteome profile,” “signal transduction,” and “gene regulatory mechanism” were used to search. The combination of these key words was also used. Then, the articles were screened and used as references for the review.

Overview of Plant Salt-Responsive Molecular Mechanisms

When plants are exposed to salt stress, they first mount sensory mechanisms to perceive salt stimuli. Hyperosmotic stress and Na+ ion toxicity are two evident components observed under salinity conditions. The root is the main organ for sensing salt stress. The sensing process is mediated by plasma-membrane and cytoplasmic proteins, G protein, Ca2+ binding protein, phosphoproteins, and ethylene receptors (Ghosh & Xu, 2014). Many aspects of sensing salt stress in plants have remained elusive. However, studies suggested that these sensors are probably associated with the mechanically-gated Ca2+ channel for hyperosmotic sensing and salt overly sensitive 1 (SOS1) Na+/H+ antiporter for Na+ sensing (Kurusu et al., 2013; Shi et al., 2000). It is believed that Ca2+ and reactive oxygen species (ROS) are produced as consequences of salinity, acting as secondary messengers. In this context, annexins have been reported to act as mediators for sensing both salt-induced high levels of Ca2+ and ROSs (Laohavisit et al., 2010).

After sensing, the messages are transduced to the downstream proteins, kinase proteins, and finally to transcription factors (Boudsocq & Sheen, 2013; Weinl & Kudla, 2009). Signaling pathways in plants include hormone pathways (abscisic acid, jasmonic acid, and ethylene), IP3 signaling pathway, and Ca2+ pathway. These signaling pathways associate with each other to transmit the stress signals to gene regulators (Zhu, 2016). Cell signaling in plants mainly depends on the SnRK family of kinases (Hardie, Schaffer & Brunet, 2016). SnRK1 regulates metabolism, SnRK2s are involved in osmotic stress and ABA signaling, and SnRK3s mediate signaling ion hemostasis (Hrabak et al., 2003). Many other proteins such as MAPK (Mitogen-Activated Protein Kinase), PK (PK Kinase), JIP (JNK-Interacting Protein), HK-ATPase (Hydrogen Potassium ATPase), WCP (Water Channel Protein), IPK (Inositol Polyphosphate Kinase), CaM (Calmodulin), CBP (Calcium-Binding Protein), and ABC transports have been identified as participating in different signaling pathways (Zhu, 2016). Many components of signal transduction, their exact functions under different types of stimuli, and their links with tolerance mechanism are still unclear.

An alarming presence of stress in the environment leads plants to organize multilevel regulatory processes in order to mount an appropriate response (Haak et al., 2017). At transcriptional level, the transcription factors, MYC2, AREB, and NAC, have been found to participate in responding to salt stress (Maruyama et al., 2009; Urano et al., 2009). Transcription factors directly bind to the promoter of genes and change the expression of many genes in response to external or internal stimuli, such as drought, salinity, and production of a specific hormone.

Alternative splicing (AS) is another regulatory level that has been detected in plants under salinity. More than 2000 AS events have been reported for salinity-treated plants (Ding et al., 2014; Laloum, Martín & Duque, 2017). Other regulatory levels reported under salinity include epigenetic modifications (Chinnusamy & Zhu, 2009; Dhar et al., 2014; Guangyuan et al., 2007; Labra et al., 2004) and miRNAs (Jian et al., 2016; Noman et al., 2017; Stief et al., 2014; Zhang, 2015). The differentially expressed genes (DEGs) are transcripted and translated to proteins and, at this level, undergo other regulational, post-translational modifications (PTMs). PTMs regulate protein function, subcellular localization, correct folding, and stability (Gong, Hu & Wang, 2015; Wu et al., 2016). The studies on PTMs of stressed plants show that the protein phosphorylation and glycosylation are accelerated under stressful conditions (Hu et al., 2013; Mustafa & Komatsu, 2014). Moreover, PTMs have commonly been observed in gel-based proteomic studies, in which one protein presents in more than one locations (spots) on the gel (Bandehagh et al., 2011; Gharelo-Shokri et al., 2016; Mann & Jensen, 2003). In the gel-based proteomics, such as sodium dodecyl sulfate-polyacrylamide gel electrophoresis (SDS-PAGE), the total proteins of samples are separated based on the mass-to-charge ratio. PTMs change this ratio. Thus, the observation that one proteins has different locations (different mass-to-charge ratio) on the gel attribute, at least in some cases, to PTMs (Bandehagh et al., 2011; Gharelo-Shokri et al., 2016; Mann & Jensen, 2003). All of these events lead to alteration of the protein abundance in different biological processes and consequently changes of cellular and molecular events in the cell.

Different tissues may have different responses under stress. In roots, changes occur mainly in proteins related to carbohydrate and energy metabolism, ion hemostasis, membrane trafficking, ROS scavenging, and cytoskeleton reorganization. In leaves, proteins, related to photosynthesis, undergo major expression changes (Deinlein et al., 2014; Ghosh & Xu, 2014; Zhang et al., 2011). It is obvious from the results of proteomic analysis reported for different organs of plants that the abundance of the responsive proteins in functional categories and biological processes, as well as their expression pattern, differ between roots and leaves (Chi et al., 2010; Gharelo-Shokri et al., 2016; Wang et al., 2014). This reveals important roles of organ-specific studies, which can have an important role in resolving the salt-mechanism enigma.

So far, thousands of salt-responsive proteins in different plants have been identified using proteomic methods. There are two important points about these proteins. First, their expression patterns are widely upregulated and only small fraction show downregulation. Second, the expression pattern and abundance of the proteins in the cellular and molecular process in the cell vary significantly from one plant to another (Zhang et al., 2011). For example, Arabidopsis thaliana induces most of the photosynthesis-related proteins when exposed to salt stress, while Thellungiella halophila reduces photosynthesis-related proteins (Pang et al., 2010). In the case of the cellular and molecular process, the majority of salt-response proteins in dicotyledonous halophytes are involved in photosynthesis, energy metabolism, ROS scavenging, and ion hemostasis (Katz et al., 2007). On the other hand, in monocotyledonous halophytes, metabolism/defense-related proteins, amino acid and TCA-related proteins, and decreased photosynthesis are main responses to salt stress (Sobhanian et al., 2010). Interestingly, both plant groups are successfully salt-tolerant, showing the importance of focused studies on the specific plant.

The Molecular Response of Canola to Salt Stress

Signal transduction

In canola roots, it is proposed that Ras-related small GTP-binding proteins mediate salt stress signaling. This protein has been identified by proteomic analysis of canola cultivars under salt stress (Banaei-Asl et al., 2015). The Ras-related small GTP-binding protein has also been identified in canola leaves under salt stress (Banaei-Asl et al., 2016; Bandehagh et al., 2011). It has previously been indicated that this protein acts as a signaling molecule, responding to salt stress, and is associated with other proteins such as G-protein-couples receptors (GPCRs) (Vernoud et al., 2003). The identification of Ras-related small GTP-binding protein which is upregulated in response to salt stress in canola (Banaei-Asl et al., 2015) may, in turn, imply a high probability of G-protein-couples receptors (GPCRs) involvement in sensing salinity signals. It has clearly been indicated that GPCRs in association with G-proteins activate Ras-related small GTP-binding protein (Bhattacharya, Babwah & Ferguson, 2004). This process is followed by activation of IP3 signaling pathway, Ca2+ production, Ca2+ pathway activation, and finally gene expression changes (Ghosh & Xu, 2014). In conjunction with the IP3 pathway role in canola response to salt stress, it has been reported that high salinity induces some components of the IP3 pathway. Transcriptomic analysis of the Brassica napus revealed that phosphatidylinositol-specific phospholipase C2 (BnPLC2), phosphatidylinositol 3-kinase (BnVPS34) and phosphatidylinositol synthase (BnPtdIns S1) have significantly differential expression under salt stress (Das et al., 2005). In the case of the Ca2+ pathway, annexin identification in canola root (Banaei-Asl et al., 2015; Yíldíz, Akçalí & Terzi, 2015) supports these pathway roles in sensing and signaling salt stress. The annexin mediator roles have been characterized in response to abiotic stresses as targets of the Ca2+ signaling pathway (Konopka-Postupolska et al., 2009). Further confirmation of the active role of the above-mentioned pathways in canola comes from identification of calcium-dependent protein kinase (CPK) differential expression under abiotic stresses, including salt stress (Zhang et al., 2014a). CPKs sense Ca2+ and act as a kinase.

Taken together, GPCRs and Ras-related small GTP-binding proteins are involved in salt stress perception, sending the message through the IP3 signaling pathway. This is followed by high accumulation of Ca2+ ions in the cytosol, mainly through the action of calcium channels located at the surface of smooth endoplasmic reticulum. In response to high Ca2+ concentration, the Ca2+ pathway is activated and, subsequently, the salinity signals are passed to the nucleus where the alteration of the relevant genes occurs in response to the salinity signals.

Gene expression regulation

Three layers of gene expression regulation have been revealed in response to salinity in canola cultivars. However, many other mechanisms remain to be explored. The first level of gene expression regulation is at the transcription stage, which is mediated by transcription factors. Transcription factors are major players interacting with other proteins, especially RNA polymerases, and cis/trans acting elements on the regulatory regions of the genome. Lee et al. (2008) reported that 56 genes in canola encode putative transcription factors as factors which are altered under abiotic stresses. Among these genes, those that have been shown upregulation by more than 5-fold under salt stress, are from AP2-EREBP family (ATERF11, CBF4/DREB1D, CBF1/DREB1B, ATERF4/RAP2.5, DREB2A, CBF1/DREB1B, DREB2A, and ATERF11), Basic-Helix-Loop-Helix (bHLH) family (AtbHLH17), Basic region leucine zipper (bZIP) family (AtbZIP55/GBF3), C2H2 family (ZAT10, ZAT12/RHL41, ZAT6, and ZAT102/RHL41), Heat stress family (ATHSFA1E), Homeobox family (ATHB-7), NAC family (ANAC036, ANAC029/ATNAP, ANAC055/ATNAC3, ANAC047, ANAC072/RD26, ANAC002/ATAF1, ANAC019, and ANAC032), and WRKY family (ATWRKY53, ATWRKY40, and ATWRKY33). These transcription factors are induced in response to the salt-stress message transmitted by sensing and signaling molecules. Thereafter, complex gene regulatory networks consisting of transcription factors and other proteins govern the expression of numerous genes.

Epigenetic events are another mechanism for gene regulation that has been shown in canola. Epigenetic regulation of stress-responsive genes has been shown to play a pivotal role in the plant under various conditions (Chinnusamy & Zhu, 2009; Luo et al., 2012). In this context, it has been reported that when salinity is added to pretreated plants with osmotic stress, plants with histone modifications accumulated Na+ ion in a concentration that is not toxic for the plant (Sokol et al., 2007). In canola, DNA methylation and histone modification have been reported in response to salinity. When canola is exposed to salt stress, de novo methylation and demethylation events occur at CpCpGpG sites (Labra et al., 2004). The genes with epigenetic modifications are less known in canola. The ethylene-responsive element binding factor (EBF) is one of the genes that undergo DNA methylation in canola under salt stress (Guangyuan et al., 2007). In this regard, studies on canola are very scarce. In Arabidopsis and tobacco, it has been shown that, under salinity, histone proteins are rapidly upregulated and are phosphorylated, resulting in a low Na+ accumulation (Sokol et al., 2007). These results suggest the possible roles of DNA methylation/demethylation and chromatin (histone) modifications in regulating salt-responsive gene expression.

In the post-transcriptional stage, Micro RNA (miRNA) roles have been studied in salt-treated canola. The miRNAs are non-coding RNA ranging from 20 to 24 nucleotides in length. It has been reported that more than 340 miRNAs participate in the post-transcriptional regulation of the salt-responsive genes in canola (Jian et al., 2016). The miRNAs are negative regulators that bind their target gene transcript and prevent the gene from being translated. It has been indicated that miRNAs induced under a specific stress, mainly target transcription factors. One of the transcription factors that are demonstrated as being targeted by miRNAs, is the NAC transcription factor (Nakashima et al., 2012). Lee et al. (2008) reported that all the transcription factors belonging to NAC family show downregulation in canola exposed to salinity. Salt tolerance homolog2 (STH2) is another target of miRNAs under salt stress reported in canola (Lee et al., 2008).

Unfortunately, there are not studies enough to indicate comprehensive information about miRNAs and their targets under salinity stress in canola. However, studies on other plants under salinity stress, specifically Arabidopsis, have revealed that many transcription factors and genes, such as superoxide dismutase and Laccases multi-copper-containing glycoproteins, are under the control of miRNAs (Liang et al., 2006; Lu et al., 2013; Stief et al., 2014). The miRNA key roles in adaptation to different types of stresses and their implications in the plant growth and development have turned attention to the use of miRNAs as a new promising target for improving tolerance to harsh environments (Zhang, 2015).

Protein synthesis and modifications

The function of the protein synthesis machinery is to supply proteins needed for the cellular processes. The proper function of this machine is vital for plants under any kind of stress. Proteomic analysis of canola roots and leaves under salinity has identified several proteins related to protein synthesis and modifications. Generally, proteins are categorized into two group: the first consists of proteins related to the proteins synthesis machine, and the second of proteins related to the correct folding and stability of newly synthesized proteins. From the first group, ribosomal related proteins (only in leaves) were identified. Ribosomal proteins constitute the ribosome structure, functioning in translation. From the second group, Heat Shock Protein (HSP) families and ubiquitin protein (only in the leaf) were identified in canola (Banaei-Asl et al., 2015; Banaei-Asl et al., 2016; Bandehagh et al., 2011; Gharelo-Shokri et al., 2016; Yíldíz, Akçalí & Terzi, 2015). Decreased protein synthesis is a common observation under salt stress (Tuteja, 2007). However, studies on tolerant cultivars of canola have shown that protein synthesis was enhanced under salinity stress, especially for ribosomal proteins, heat shock proteins, and ubiquitin proteins (Banaei-Asl et al., 2015; Banaei-Asl et al., 2016; Bandehagh et al., 2011). It seems that canola’s strategies in the root and leaves are different. In the root, only the differential expression of heat shock proteins, Hsp 70, has been reported (Banaei-Asl et al., 2016; Bandehagh et al., 2011), while in the leaves, ribosomal, heat shock, and ubiquitin proteins have been upregulated (Banaei-Asl et al., 2015). Hsp 70 is a chaperone protecting newly synthesized proteins from aggregation as well as ensuring proper folding. Their high accumulation is induced by many environmental stresses, such as heat, drought, salinity, cold, and wound healing (Boston, Viitanen & Vierling, 1996; Burdon, 1988). Similar to Hsp 70, ubiquitin acts as a regulator to stabilize the functions and location of the proteins through covalently binding those proteins at specific sites (Dametto et al., 2015).

Dynamic changes of canola genes and proteins

Several transcriptomic and proteomic studies performed on canola under salt stress indicating that DEGs and differentially expressed proteins in both leaves and roots are mainly categorized into seven functional groups, except genes/proteins related to signaling, transcription, protein synthesis and modifications described in the earlier sections (Banaei-Asl et al., 2015; Banaei-Asl et al., 2016; Bandehagh et al., 2011; Deinlein et al., 2014; Gharelo-Shokri et al., 2016; Jia et al., 2015; Yíldíz, Akçalí & Terzi, 2015). According to the number of genes/proteins identified in each functional group, these groups are (a) carbohydrate and energy metabolism, (b) stress and defense, (c, d) metabolism and photosynthesis (in the case of leaves), (e) cell structure, (f) membrane and transport, and (g) cell division, differentiation and fate. These groups have different members in number but they are respectively mentioned from high to low number (Deinlein et al., 2014). In canola roots subjected to stress, the number of proteins related to carbohydrate and energy metabolism are more than in other functional groups. However, proteins related to amino acid metabolism and cell structure are also remarkable in abundance. In carbohydrate and energy metabolism the majority of proteins are from the TCA cycle, electron transport chain (ETC), and glycolysis (Banaei-Asl et al., 2015; Gharelo & Bandehagh, 2017). In canola leaves, the high abundance functional proteins belong to photosynthesis, protein synthesis and degradation, amino acid metabolism, and damage repair and defense response (stress and defense) (Banaei-Asl et al., 2016; Gharelo-Shokri et al., 2016; Jia et al., 2015). In the photosynthesis-related, differential abundance of chlorophyll a/b binding protein, chloroplast RuBisCO activase, ribulose bisphosphate carboxylase (RuBisCO) small and large subunit, and ribulose bisphosphate carboxylase/oxygenase have been reported in salinity-tolerant canola cultivars (Banaei-Asl et al., 2016; Bandehagh et al., 2011; Yíldíz, Akçalí & Terzi, 2015). Because of the importance of membrane transporter proteins, stress and defense, and amino acid metabolism for salt tolerance, we described the proteins and genes identified in these functional groups at forthcoming in a subsequent section in detail. Here, it is worth mentioning two frequently observed proteins, actin, and tubulin, identified in canola root and leaves (Banaei-Asl et al., 2016; Bandehagh et al., 2011; Jia et al., 2015; Yíldíz, Akçalí & Terzi, 2015). Similar to other plant response to salt conditions (Agarwal et al., 2013; Bandehagh et al., 2011; Gharelo-Shokri et al., 2016; Ghosh & Xu, 2014; Gupta & Huang, 2014; Liang et al., 2018; Mickelbart, Hasegawa & Bailey-Serres, 2015; Parihar et al., 2015; Tuteja, 2007; Wan et al., 2017; Yíldíz, Akçalí & Terzi, 2015; Zhang et al., 2011), it seems that canola alters its cytoskeleton basic components (i.e., actin and tubulin) under salt stress. It has been demonstrated that cytoskeleton dynamic remolding is linked to some of the main transmembrane transports, such as K+ channel (Ahanger et al., 2017; Katz et al., 2007; Tuteja, 2007). Another important point relating to the functional category of differentially changed proteins is the unknown proteins that constitute about 1% to 20% of total differentially changed proteins in each study results, especially in studies on the root (Banaei-Asl et al., 2015; Banaei-Asl et al., 2016; Bandehagh et al., 2011; Gharelo-Shokri et al., 2016; Jia et al., 2015; Toorchi & Kholgi, 2014; Yíldíz, Akçalí & Terzi, 2015; Zhang et al., 2011). Identification of these proteins could provide more insights on salt-response mechanisms.

In each functional category, the abundance of genes/proteins dynamically changes on the basis of duration and severity of salt stress, organ, and even between different leaves. Bandehagh et al. (2011) reported that when salinity severity increases from 175 to 350 mM NaCl, the number of differentially expressed proteins increase. They compared the increases between second leaf (young leaves) and third leaf (older leaves) and conclude that the younger leaves show significantly more increase in their salt-responsive proteins. Another study, by Hu et al. (2013), indicated that the expression levels of BnBDC1, BnLEA4, BnMPK3, and BnNAC2 are upregulated 1 h after salt stress, while their expression is downregulated 3, 6, and 12 h after the stress. Jia et al. (2015) in their study on the dynamic changes of canola’s proteome under 200 mM NaCl at three-time points (24 h, 48 h, and 72 h) indicated that the salt-responsive proteins have a dynamic pattern. They explained the dynamic behavior of canola proteome by clustering the salt-responsive proteins into two main clusters. In the first cluster, the proteins were grouped into two sub-groups, A and B. The sub-group A showed downregulation at 24 h after salinity treatment, while these proteins (sub-group A) were upregulated at 48 h and 72 h time points. These studies indicate the importance of dynamic analysis in understanding molecular mechanisms. Investigation of the different organs, different time points, and different severities of salt stress as well as integrated transcriptomic and proteomic dynamics could provide deep insights into the response of canola to salinity.

Canola Molecular Salt-Tolerance Mechanisms

Engineering of salt-tolerant crops has been a long-standing objective. Although, in some cases, crops transformed with a certain gene, treated with an exogenous material, or inoculated with specific strains of bacteria have shown more tolerance to salt stress, this tolerance was not at a level that could be practiced in the real field conditions. Studies of tolerant plants have identified several common or specific salt-tolerance mechanisms. Halotropism is one recently defined mechanism in which the plant alters its root growth pattern. In this type of response, salt-induced auxin changes the root growth direction to avoid highly saline media (Galvan-Ampudia et al., 2013). In another mechanism, the plants increase K+/Na+ ratio which is defined as a key salt tolerance trait. It has been indicated that endosomal Na+/H+ antiporter, plasma-membrane located SOS transports, and H+/K+ transporters are basic players in this regard. These membrane proteins confer the plant Na+detoxification ability from the cytosol and selective absorption of K+ions (Liang et al., 2018; Wan et al., 2017). The salt-tolerant plant, especially in early stages of exposure to salinity media, accumulates soluble sugars, proline, glycine betaine, and other osmolytes. In connection with this mechanism, P5CS (Pyrroline-5-carboxylate synthase), rate-limiting enzyme of proline metabolism, has well been studied by knocking out or overexpression methods. Studies confirmed the correlation of overexpression of the enzyme with high salinity tolerance and vice versa (Hmida-Sayari et al., 2005; Hur et al., 2004; Zhu et al., 1998). A further salt tolerance mechanism is the ROS scavenging system comprising glutathione ascorbate pathway, CAT pathway, PrxR/Trx pathway, and GPX pathway. The enzymes superoxide dismutase (SOD), glutathione peroxidase (GPX), ascorbate peroxidase (APX), and catalase (CAT) are most important antioxidants in this system. Increasing activities of these enzymes and overexpression of their encoding genes significantly improve plant salt tolerance (Wang et al., 2016a; Wang et al., 2016b). With this background, we review some canola salt tolerance molecular components that have been identified in the root and leaves using comparative analysis between tolerant and sensitive cultivars (Fig. 1).

Figure 1 Schematic representation of the major salt-tolerant components identified in canola by proteomic, transcriptomic, and genetic engineering methods.

+, upregulation; −, downregulation; upward red arrows, high activity. Abbreviations: 3-PGA, Glycerate 3-phosphate; ACAs, Calcium-transporting ATPase; AP2-EREBP, AP2-like ethylene-responsive transcription factor; APX, Ascorbate peroxidase bHLH, Helix-loop-helix transcription factor family bZIP, Basic leucine zipper transcription factor family; CAT, Catalase; CAX2, Vacuolar cation/proton exchanger2; CNGCs, Cyclic nucleotide-gated ion channels; CPKs, Calcium-dependent protein kinases; DREBs, Dehydration-responsive element-binding proteins; FBA, Fructose-bisphosphate aldolase; GAPDH, Glyceraldehyde-3-phosphate dehydrogenase; GLR, Glutamate receptor; GPCRs, G-protein coupled receptors; GPX, Glutathione peroxidase; GSA/P5C, Glutamate-1-semialdehyde 2,1-aminomutase/Pyrroline-5-carboxylate; HKT1, Sodium transporter HKT1; IP3, Inositol 1,4,5-trisphosphate; KCO6, Two-pore potassium channel 3; KEAs, K+ efflux antiporters; MDH, Malate dehydrogenase; NAC, NAC domain-containing proteins; OAT, Ornithine aminotransferase; P5CR, Pyrroline-5-carboxylate reductase; P5CS, Pyrroline-5-carboxylate synthase; PHT1.4, Inorganic phosphate transporter 1-4; PLC2, Phosphoinositide phospholipase C 2; PO, Proline oxidase; PtdinsS1, phosphatidylinositol synthase; Rubisco, Ribulose bisphosphate carboxylase; RuBP, Ribose 1,5-bisphosphate; SAM, S-adenosylmethionine; SAMS, S-adenosylmethionine synthetase; SDH, Succinate dehydrogenase; SOD, superoxide dismutase; SOD1, Cu/Zn superoxide dismutase; SOD2, Mn-superoxide dismutase; TIM, Triosephosphate isomerase; V-ATPase, V-type proton ATPase; WRKY, WRKY transcription factor.

Proline synthesis

Studies on canola have indicated significantly increased proline contents in the root and leaves of both salt-tolerant and sensitive cultivars under salt stress, and comparatively more in tolerant cultivars than sensitive ones (Dolatabadian, Sanavy & Chashmi, 2008; Xue, Liu & Hua, 2009). Proteomic analysis of canola-tolerant cultivars indicated that the abundance of P5CS in the root and leaves are increased (Banaei-Asl et al., 2015; Banaei-Asl et al., 2016; Yíldíz, Akçalí & Terzi, 2015). It has been shown that high accumulation of proline in canola attributes to activating its biosynthesis and preventing its degradation (Xue, Liu & Hua, 2009). This was confirmed in another study by Madan et al. (1995) in which they reported that the activity of proline synthesis enzymes, pyrroline-5-carboxylate reductase and ornithine aminotransferase, increases about three-fold, whereas the proline-degrading enzyme, proline oxidase, was decreased by salt stress. These findings confirm that canola changes proline metabolism in tune with the increase of proline contents under salt stress. Proline acts as an osmolyte, ROS scavenger, redox buffer, and molecular chaperones under stressful conditions. This suggested that, during the recovery phase, proline serves as a signaling molecule for cell growth, proliferation, and death (Deinlein et al., 2014; Liang et al., 2018; Xue, Liu & Hua, 2009).

Alternative regulation of ion transporter proteins in the root and leaf

H+/K+ transporters (HKT) and Na+/K+ co-transport show downregulation in canola root but upregulation in the leaves during salt stress (Yong et al., 2014). This pattern of HKT expression is to reduce Na+ uptake through the root from rhizosphere and to protect the leaf against Na+ accumulation by removing Na+ from xylem sap into parenchymal cells. These type of ion transporters are responsible for Na+ uptake into the root and leaf cells. In Arabidopsis, it has been reported that mutants with HKT deficiency are salt hypersensitive with a high amount of Na+ in leaves and a low amount in the root (Mäser et al., 2002). Furthermore, the role of HKTs has been indicated in removing Na+ ions from xylem sap into parenchymal cells located around xylem (Berthomieu et al., 2003).

Three transporters involved in transporting Ca2+ into the cytosol, including CNGCs (cyclic nucleotide-gated ion channel), GLRs (glutamate receptor), and ACAs (ATPase homologous to Arabidopsis), have been shown to be upregulated both in canola root and leaf, while CAXs (endosomal cation/proton exchanger), which is involved in removing Ca2+ from the cytosol into the vacuole, is downregulated in the root (Yong et al., 2014). It seems that overall actions of these ion transporters are to increase Ca2+ ions in the cytosol. High levels of Ca2+ could have two main consequences for the cell. First, Ca2+ ions accumulated in the cytosol activate Ca2+ sensor proteins of the cytosol to signal the presence of stress (Knight, Trewavas & Knight, 1997). Second, these ion channel actions in transporting Ca2+ ions also contribute to K+ uptake which is important for maintaining the cellular hemostasis (Ali, Zielinski & Berkowitz, 2005).

Canola upregulates ion channels responsible for K+ influx into the cytosol from vacuole (KCOs) and from stele into xylem (SKORs) (Labra et al., 2004). However, it downregulates K+ ion channels functioning in the efflux of K+ out of the cell (KEA) (Yong et al., 2014). It has been illustrated that K+ functions in controlling whole plant ion hemostasis and the cell turgor (Shabala & Cuin, 2008; Su et al., 2001). In tune with these changes in K+ transporters, PHT and V-ATPase are upregulated by canola (Yong et al., 2014). These proteins act to provide the driving force to maintain Na+ at a low level and K+ in a high level in the cell cytosol (Zhu, 2003).

Reactive oxygen species scavenging system

For ROS production, there are two conflicting views: while many reports blame ROSs for attacking major components of the cell and for interfering with many vital metabolic pathways, some reports do not know ROS production necessarily as a symptom of these dysfunctions, but place emphasis on ROS signaling roles under abiotic stress. However, all studies have found the efficiency of the plant ROS scavenging system to be one of the main stress-tolerant traits. Regarding this, a significantly high activity of APX and CAT is reported for a canola-tolerant cultivar (i.e., Hyola308) treated with 200 and 300 mM NaCl (Heidari, 2010). Similarly, another study reported high activity of SOD, GPX, and CAT as well as upregulation of Cu/Zn SOD in the leaves under 150 mM NaCl. Upregulation of Cu/Zn SOD has similarly been demonstrated for Hyola308 under 150 and 300 mM NaCl stress (Banaei-Asl et al., 2016). All these studies provide evidence to confirm that tolerant cultivars show remarkably more ROS scavenging activity than sensitive cultivars (Banaei-Asl et al., 2016; Heidari, 2010; Yíldíz, Akçalí & Terzi, 2015). Furthermore, the role of glutathione synthesis in canola is defined as oxidative protective mechanism (Ruiz & Blumwald, 2002). In this context, cysteine and glutathione content was measured for wild-type and salt-tolerant transgenic canola, transformed with vacuolar Na+/H+ antiporter from Arabidopsis, under 150 mM NaCl. After 15 days of continuous salt stress, cysteine and glutathione content were increased three-fold in wild compared to tolerant canola. This observation confirms the possible protective role of glutathione synthesis in coping with oxidative damages.

Candidate genes/proteins for canola tolerance

Several studies have tried to identify the main gene(s)/protein(s) responsible for canola tolerance. Knowledge of these key components among the complexity of the salt response networks is a critical step toward engineering salt-tolerant canola. Gharelo-Shokri et al. (2016) in their study reported six hub genes in tolerant cultivars, including UDP-glucose dehydrogenase, Methionine synthase, Malate dehydrogenase, Triose phosphate isomerase, heat shock protein 70, and Fructose-bisphosphate aldolase in constructed protein-protein interaction network of canola salt-induced proteins. Hub genes are high interactive elements of the constructed network that is regarded as the main components of the network (Ning et al., 2010). Furthermore, some of the candidate genes/proteins for canola salt tolerance could be extracted from studies about an external application of materials that promote canola tolerance under salinity. In this respect, Banaei-Asl et al. (2015) found that, in response to plant growth promoting rhizobaceria (PGPR) inoculation, canola root upregulates glyceraldehyde-3-phosphate dehydrogenase and downregulates S-adenosylmethionine synthase, aldehyde dehydrogenase, and malate dehydrogenase under 150 and 300 mM of NaCl. Their study demonstrated that inoculated plants show significantly more root length, root dry weight, high K+ levels, and a low Na+ and Cl− levels compared to non-inoculated plants. They showed that the better growth parameters of the inoculated root are due to differential abundant bacteria-responsive proteins. Moreover, in another study, they indicated that bacterial inoculation gives canola more tolerance through an increased abundance of proteins related to glycolysis, TCA, and amino acid metabolism, especially succinate dehydrogenase (Banaei-Asl et al., 2016).

In canola, several studies have shown that overexpression of some genes results in altered salt tolerance. One example is ectopic overexpression of Dehydration-Responsive Element Binding transcription factors (DREBs). Plants transformed for high expression of DREBs increase their salt-responsive gene expression including COR14, HSF3, HSP70, PEROX and RD20 showing more tolerance. According to these results, transgenic plants are able to survive under the salinity level in which wild-type plants are more sensitive (Shafeinie et al., 2014). Given that 5-Aminolevulinic acid (5-ALA) exogenous application resulted in salt tolerance in treated plants, Sun et al. (2015) transformed canola with 5-ALA-encoding gene, YHem1, and studied the growth of the transgenic canola capable to synthesize more 5-ALA, and wild-type canola under salinity stress. They reported that under both short-term and long-term salinity, transgenic canola show more yield, more chlorophyll content, high activity of antioxidant enzymes, high proline content, high sugar content, and more free amino acids compared to wild-type canola. Furthermore, they demonstrated that an increased tolerance of transgenic canola could be related to the upregulation of Rubisco small subunit and significantly high levels of Fe metal. In contrast to these studies, in which increased tolerance has been reported, it has been suggested that expression of Brassica napus TTG2 causes sensitivity to salt stress through downregulation of the genes TRYPTOPHAN BIOSYNTHESIS 5 (TRP5) and YUCCA2 (YUC2) encoding indole-3-acetic acid (IAA), hence, declining the endogenous IAA content. It is expected that in future research in transgenic plants, the newly emerging CRISPR/Cas9 system will provide more information about molecular components responding to salt stress (Osakabe et al., 2016).

Concluding Remarks and Outlook

Canola, one of the most important field crops in the world, is affected by salt stress. In spite of advances in understanding the plant-salt molecular interactions, developing salt-tolerant varieties remains challenging. According to the studies reviewed in this report, the molecular components of the salt-response network vary among plants, species, organs, and tissues. Focused and complementary studies with integrated approaches are therefore essential to identify the key elements for use in plant genetic engineering. This report examines the results of proteomic, transcriptomic, genetic engineering, and genetic studies to put together current understanding of canola salt response. However, many aspects of this response remain unknown and the studies available only identify some of the cellular and molecular responses. The roles of DNA methylation, histone modifications, alternative splicing, miRNAs, and protein post-translational modifications are not well understood. The majority of studies in canola have been conducted on the leaf and little is known about the root and other tissues. In the proteomic studies, a portion of proteins/genes detected as salt-response components remains unidentified. At the cellular level, crosstalk between pathways is little understood. There are many unclear aspects about the effects of the interactions between different salt tolerance traits and the roles of organelle genetic material in responding to salt stress.

Progress has been made in the methods, such as free-gel-based proteomic methods, multidimensional separation of proteins, tagging of phosphoprotein and glycoprotein, high throughput DNA and RNA sequencing and quantifications technologies, CRISPR/Cas9 system, system biology, and in silico prediction/modeling of the plant salt-response network(s). This progress will remarkably help to elucidate the unknown or less-known aspects. Integration of these method results could provide knowledge of key components of the salt-response network(s), low abundant proteins, novel regulatory mechanisms, and metabolic pathways crosstalk, particularly in transcription factors and signaling molecules.

Our warmest appreciation goes to Dr. Hilary Cadman and Mr. Peter Saunders for their editing and proofreading support of the manuscript language.

Additional Information and Declarations

Competing Interests

Author Contributions

Data Availability

The authors declare there are no competing interests.

Reza Shokri-Gharelo conceived and designed the experiments, performed the experiments, analyzed the data, contributed reagents/materials/analysis tools, prepared figures and/or tables, authored or reviewed drafts of the paper, approved the final draft.

Pouya Motie Noparvar authored or reviewed drafts of the paper, approved the final draft.

The following information was supplied regarding data availability:

The research in this article did not generate any data or code. This article is a literature review.

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
