# Peer review of "Molecular response of canola to salt stress: insights on tolerance mechanisms"

_PeerJ, doi:10.7717/peerj.4822_

## Round 0.1 · original submission · Minor Revisions

Based on the unanimous opinion of 3 reviewers and my own reading, it is clear that you are presenting an interesting and important review, which is generally well-written, is of broad interest, and will be a useful addition to the field.

Some minor issues are raised by two reviewers and obviously they need to be addressed. Please pay close attention to the reviewer #3 request to provide a clear description of different techniques you are discussing in your article. Probably, a good idea would be to have a special section, in which various techniques are introduced and their strengths and weaknesses are discussed.

·

Basic reporting

1. Line 30, can rephrase to "suggests what researchers should focus on in future studies".
2. Line 45, "are taken up" to "take up" or "absorb".
3. Line 46, "are" to "is"; "compound" to "compounds".
4. Line 56, delete the 2nd "is"; Line 57, delete the "is".
5. Line 64-66, references are needed here.
6. Line 81-87, the Survey Methodology section is not needed.
7. Line 115-116, the sentence "Transcription factors directly change the expression level of many genes" is unclear.
8. Line 127-132, the SDS PAGE principle is not necessary; it is just a basic analytic technique.
9. Line 161, for a literature review, "seems that" sounds unsure, can be rephrased to it is proposed that.
10. Line 165, the 1st sentence can be deleted.
11. Line 199-200 can be deleted. The basic molecular biology concept is not necessary here.
12. Line 211-212, the sentence is unclear.
13. Line 220, "less" than what?
14. Line 221, "undergoes" to "undergo".
15. Line 227, "post-translational" to "post-transcriptional".
16. Line 254-255, this sentence is not needed.
17. Line 272-278, references are needed.
18. Line 294, "future" to "subsequent".
19. Line 310-312, this sentence is unclear. a number of or the number of?
20. Line 322, what "these protein" represents? (Sub-group A was mentioned).
21. Line 338, "membranous" to "membrane".

Experimental design

no comment

Validity of the findings

no comment

·

Basic reporting

English is clear and professional.

Experimental design

Experiment design looks good and research is under scope of the journal

Validity of the findings

Experiments are well designed.

Additional comments

i recommend acceptance of the article for publication.

Reviewer 3 ·

Basic reporting

In this review, the authors focused on studies of the molecular response of canola to salinity to unravel the different pieces of the salt response puzzle. The paper includes a comprehensive review of the latest studies, particularly of proteomic and transcriptomic analysis, which will be helpful for understanding the mechanisms of salt tolerance and cultivating the salt tolerant canola varieties. However, this manuscript still has some content need to be modified or explanted.

Experimental design

The experimental design is reasonable。

Validity of the findings

no comment

Additional comments

1. Line 468-470: the authors introduced that the report examines the results of proteomic, transcriptomic, genetic engineering, QTL mapping, and genetic studies to put together current understanding of canola salt response. However, there is no description of QTL mapping in the text, and there is less explanation in transcriptome. Could you please make an explanation?
2. line 479-486: in this paragraph, the author introduced many methods, and told us that the progress of these methods will remarkably help to elucidate the unknown or less-known aspects, whether the author can explain in detail how they can be applied in future studies.
3. In the paragraph of ‘Dynamic Changes of Canola Transcriptome and Proteome. This section describes the transcriptome dynamic change rarely, only discussed in proteomics level. Could you please add some content about dynamic changes of canola transcriptome?
4. line142 ‘root and leaves’ was changed to ‘roots and leaves’.
5. line250 ‘root and leaves’ was changed to ‘roots and leaves’.

---

## Round 0.2 · accepted · Accept

Thank you very much for your response to the reviewers' comments and for the careful revision of the manuscript. I agree with your decision to keep "Survey Methodology" section in the manuscript.

#